# Dual-Band MIMO Antenna for 5G/WLAN Mobile Terminals

**DOI:** 10.3390/mi12050489

**Published:** 2021-04-26

**Authors:** Jianlin Huang, Guiting Dong, Qibo Cai, Zhizhou Chen, Limin Li, Gui Liu

**Affiliations:** College of Electrical and Electronics Engineering, South Campus, Wenzhou University, Wenzhou 325006, China; 194511981414@stu.wzu.edu.cn (J.H.); 20451941039@stu.wzu.edu.cn (G.D.); caiqibo@wzu.edu.cn (Q.C.); 20170157@wzu.edu.cn (Z.C.); lilimin@wzu.edu.cn (L.L.)

**Keywords:** 5G antenna, sub-6 GHz, MIMO, dual-band antenna, open-loop ring resonator

## Abstract

This paper presents a dual-band four-element multiple-input-multiple-output (MIMO) array for the fifth generation (5G) mobile communication. The proposed antenna is composed of an open-loop ring resonator feeding element and a T-shaped radiating element. The utilization of the open-loop ring resonator not only reduces the size of the antenna element, but also provides positive cross-coupling. The dimension of a single antenna element is 14.9 mm × 7 mm (0.27λ × 0.13λ, where λ is the wavelength of 5.5 GHz). The MIMO antenna exhibits a dual-band feature from 3.3 to 3.84 GHz and 4.61 to 5.91 GHz, which can cover 5G New Radio N78 (3.3–3.8 GHz), 5G China Band N79 (4.8–5 GHz), and IEEE 802.11 ac (5.15–5.35 GHz, 5.725–5.85 GHz). The measured total efficiency and isolation are better than 70% and 15 dB, respectively. The calculated envelope correlation coefficient (ECC) is less than 0.02. The measured results are in good agreement with the simulated results.

## 1. Introduction

Fifth generation (5G) communications technology can offer many advantages over the fourth generation (4G) systems, such as higher transmission rates and shorter latency. To obtain the increasing demands of 5G communication, the multiple-input multiple-output (MIMO) technique is a promising technology in antenna design [1,2,3]. In sub-6 GHz 5G operations, MIMO antenna systems with high isolation, high gain and low ECC are needed to achieve high transmission rate and large channel capacity.

Recently, many 5G MIMO antenna systems have been developed [4,5,6,7,8,9,10,11,12,13]. To achieve the bandwidth response, different radiators have been presented, such as inverted F-shaped [4,5,6], Yagi-shaped [7], π-shaped [8], Y-shaped [9], T-shaped [10] and folded L-shaped [11]. A folded monopole and a gap-coupled loop branch are formed in [12] to realize good mutual coupling reduction. In [13], an L-shaped strip and a modified Z-shaped strip are employed to create the lower and higher resonance frequency, respectively. A metamaterial structure is used to improve the bandwidth and reduce the mutual coupling [14].

Due to the limited space of the smartphones, the approach to design MIMO antenna array with good isolation performance is still a crucial challenge [15,16]. In many reported antennas, various methods have been proposed to improve isolation between antenna elements [17,18,19,20,21,22]. In [17], four pairs of orthogonal antennas are placed symmetrically on the corner of the smartphone to enhance isolation. A common grounding branch for adjacent antenna elements is proposed to improve isolation [18]. Two antennas are isolated in an orthogonal manner [19] which has been reported to achieve high isolation. A balanced slotted antenna is designed in [20], which can generate a balanced slotted pattern to improve the isolation performance. In [21], the antenna is composed of a double-CPW-fed antenna with a pair of modified T-ring radiators, which can also achieve antenna diversity. The antenna applies the orthogonal monopole/dipole modes in the lower band while using the orthogonal slot and open slot modes in the higher band to obtain a high isolation during the entire frequency band [22].

The third Generation Partnership Project (3GPP) has proved that 5G New Radio (NR) involves three sub-6GHz operating bands, i.e., N77 (3.3–4.2 GHz), N78 (3.3–3.8 GHz) and N79 (4.4–5.0 GHz). Diverse countries and areas can choose their own 5G operating bands from the above three bands. For example, the use of 3.3–3.6 GHz and 4.8–5.0 GHz bands has been announced officially by China, the 3.6–4.2 GHz and 4.4–4.9 GHz has been utilized globally by Japan, and the 3.4–3.8 GHz band has been adopted by European Union (EU). Designing dual-frequency compact antenna with broad bandwidth in both bands is often a challenging task for antenna designers. The loop antennas can significantly minimize the size. Usually, the 0.5λ, 1.0λ and 1.5λ modes are the first three resonance modes of loop antennas. Besides, an open-slot, a tuning section and a parasitic section at the reverse side can further boost the bandwidth for both the lower and higher bands.

In this paper, a novel ring antenna for the applications of 5G MIMO antenna system in mobile terminals is proposed. The antenna array consists of an open-loop ring resonator and a T-shape radiating element. The combination of the 0.27λ and 0.13λ modes forms an obvious dual-band operation that can cover 3.3–3.84 GHz in the lower band and 4.61–5.91 GHz in the higher band. Different radiators have been designed and analyzed for the proposed compact 4-antenna MIMO system. The capability of the proposed 4-antenna MIMO system is verified consistently by both simulation and measurement.

## 2. Antenna Geometry

The detailed geometry of the proposed dual-band four-element array for 5G MIMO terminal applications is shown in Figure 1. There are two kinds of printed circuit boards (PCBs) in the proposed MIMO antenna, including a main board and two side boards. The dimensions of the main board and the side boards are 150 mm × 75 mm × 0.8 mm and 134 mm × 6.2 mm × 0.8 mm, respectively. Four antenna elements are printed on the two side boards which are placed vertically to the main board. The side boards and main board are all printed on FR4 with εr = 4.4 and tanδ = 0.02, which are bonded to each other by metal glue. There are two clearance areas (75 mm × 8 mm) situated at both top and bottom sides of the main board, which are reserved for 2G/3G/4G or other wireless communication systems in current mobile handsets. Figure 1b shows the detail dimensions of a single antenna element that is a dual-band antenna operating in 3.3–3.84 GHz and 4.61–5.91 GHz. As shown in Figure 1, the dual-band antenna contains an open-loop ring resonator feeding element and a T-shaped radiating element which is connected to the ground plane. The total size of a single element is 14.9 mm × 7 mm × 0.8 mm, which takes advantage of the resonator to reduce the size and improve the performance of the individual element.

## 3. Antenna Analysis

In order to investigate the dual band operating mechanism, a parametric study of single antenna element has been carried out by the electromagnetic simulator HFSS. Since the geometry of the four antenna elements are identical, the parameters of Ant. 1 are used for analysis.

Figure 2 exhibits the surface current and electric field distributions of the proposed Ant. 1. The 3.5 GHz surface current is concentrated on the right side of the individual antenna element toward the center of the substrate, while the electric field is reversed toward the outside of the substrate. However, the surface current at 5.5 GHz is concentrated to the left side of the individual antenna element, while the electric field is directed toward the interior of the substrate. These two resonant positions result in dual-band and wideband characteristics.

Figure 3 shows the simulated S_11_ of Ant. 1 as functions of L_1_ and H_3_, respectively. The value of L_1_ can be effectively employed to shift the resonance frequencies of both the lower and higher frequency bands. Eventually, both the N78 (3.3–3.8 GHz) band and IEEE 802.11 ac (5.15–5.875 GHz) band can be covered. As shown in Figure 4, the higher resonance of Ant. 1 can also be optimized by the value of H_3,_ which is the length of the open-loop ring.

Figure 5 shows the simulated application scenario when Ant. 3 is excited. Due to the high permittivity and high loss characteristic of the hand, it is necessary to investigate the effect of the hand on the antenna performance.

Figure 6 shows simulated coefficients when the device is held in single hand. To cover both the N78 (3.3–3.8 GHz) band and IEEE 802.11 ac (5.15–5.875 GHz) bands, the reflection coefficients are lower than −8.5 dB. The transmission coefficients are lower than 14 dB.

Figure 7 and Figure 8 show simulated electric field distributions and 3D radiation patterns when the device is held in a single hand, respectively.

## 4. Experimental Results and Discussion

To validate the proposed design, an antenna prototype was fabricated with the optimized dimensions listed in Figure 1. Figure 9 shows the photograph of the measurement experiment, including the vector network analyzer (VNA) instrument, the microwave anechoic chamber, and the fabricated antenna prototype.

The S-parameters are measured by Keysight Vector Network Analyzer N5224A. The measured S-parameters are shown in Figure 10. The −10 dB impedance bandwidth (3.3–3.84 GHz and 4.61–6.8 GHz) can completely cover 5G New Radio N78 (3.3–3.8 GHz), 5G China Band N79 (4.8–5 GHz), and IEEE 802.11ac (5.15–5.35 GHz, 5.725–5.85 GHz). A slight frequency shift can be observed, which may be due to several factors such as soldering-related effects and limitation of the milling machine. The measured isolation between each port is better than 15 dB.

As depicted in Figure 11, the measured total efficiencies are better than 70%.

Figure 12 shows that the measured realized peak gain are better than 4.2 dBi.

Figure 13 shows the measured reflection and transmission coefficients when the device is held in a single hand. The reason for the low frequency variation is the current distribution at the 3.5 GHz towards the inner direction of the substrate, which is disturbed by hand. However, the −6 dB impedance bandwidth is still covering 3.3–3.8 GHz and the isolation is better than 14.2 dB.

Figure 14 shows that the measured reflection coefficient and transmission coefficient when holding the device with both hands. Different from single-hand holding, dual-hand holding has less effect on S-parameter in the lower frequency band. The −9 dB impedance bandwidth can cover 3.3–3.8 GHz and the isolation is better than 14.4 dB.

In this study, the imaginary part and real part of the S-parameters are measured by VNA, and the calculated ECC formula for the proposed eight-antenna MIMO system is as follows:(1)$$\rho_{\mathrm{ij}}=\frac{{\left|S_{\mathrm{ii}}^{*}S_{\mathrm{ij}}{\;+\;S}_{\mathrm{ji}}^{*}S_{\mathrm{jj}}\right|}^{2}}{(1-({\left|S_{\mathrm{ii}}\right|}^{2}+{\left|S_{\mathrm{ji}}\right|}^{2}))(1-({\left|S_{\mathrm{jj}}\right|}^{2}+{\left|S_{\mathrm{ij}}\right|}^{2}))}$$

The calculated envelope correlation coefficient (ECC) of the proposed MIMO antenna system is shown in Figure 15. The values of the ECC in the operating frequency bands are smaller than 0.02.

The radiation patterns of Ant. 1 at 3.5 GHz and 5.5 GHz are shown in Figure 16. The measured co-pol and cross-pol are represented by solid and dash lines. The left column is the xoy plane, and the right column is the yoz plane.

Table 1 shows a comparison between the presented antenna and other reported smartphone 5G MIMO antennas.

## 5. Conclusions

A four-element dual-frequency MIMO antenna system which can cover N78 (3.3–3.8 GHz), 4.9 GHz (4.8–5 GHz) and IEEE 802.11 ac (5.15–5.35 GHz, 5.725–5.85 GHz) is proposed in this paper. The total efficiency is more than 70%, and the isolation performance is better than 15 dB. The measured results are in good agreement with the simulated results. The proposed MIMO antenna could be a good candidate for both sub-6 GHz 5G and WLAN applications in mobile terminals.

## Figures and Tables

**Figure 1 micromachines-12-00489-f001:**
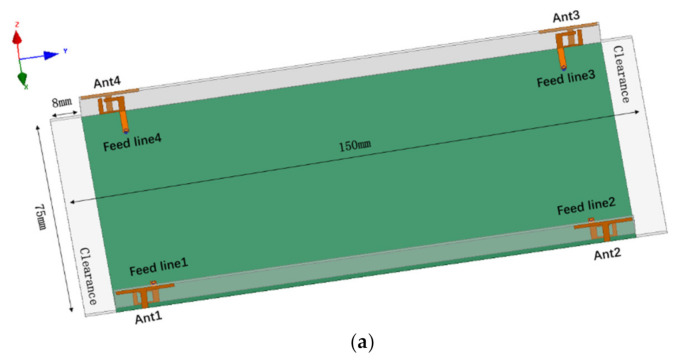
Geometry of the proposed MIMO antenna system in millimeters. (**a**) Prospective view; (**b**) single antenna element.

**Figure 2 micromachines-12-00489-f002:**
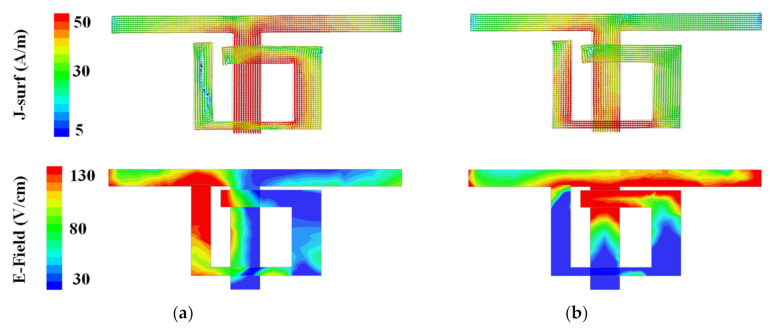
The surface current and electric field distributions of the proposed antenna at (**a**) 3.5 GHz; (**b**) 5.5 GHz.

**Figure 3 micromachines-12-00489-f003:**
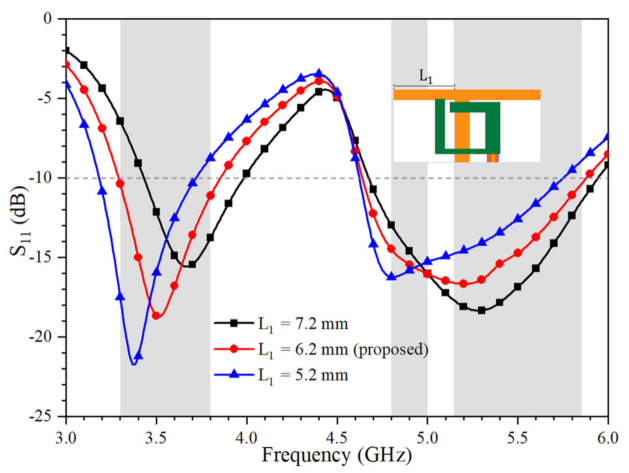
Simulated S_11_ of Ant. 1 with different values of L_1_.

**Figure 4 micromachines-12-00489-f004:**
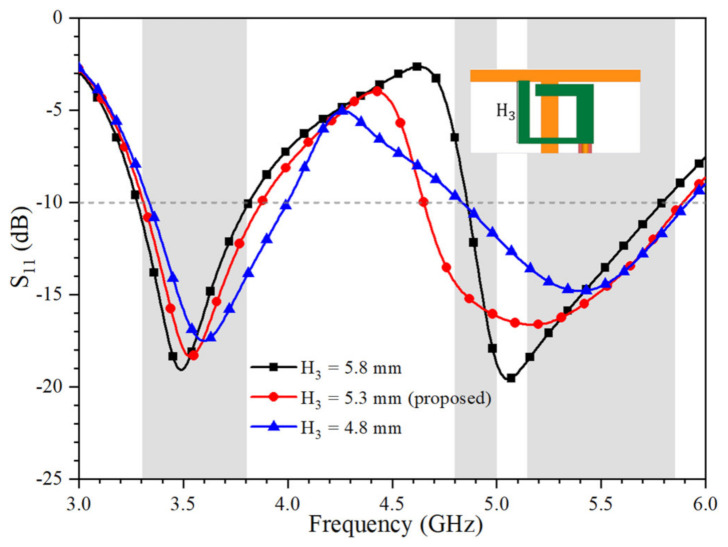
Simulated S_11_ of Ant. 1 with different values of H_3_.

**Figure 5 micromachines-12-00489-f005:**
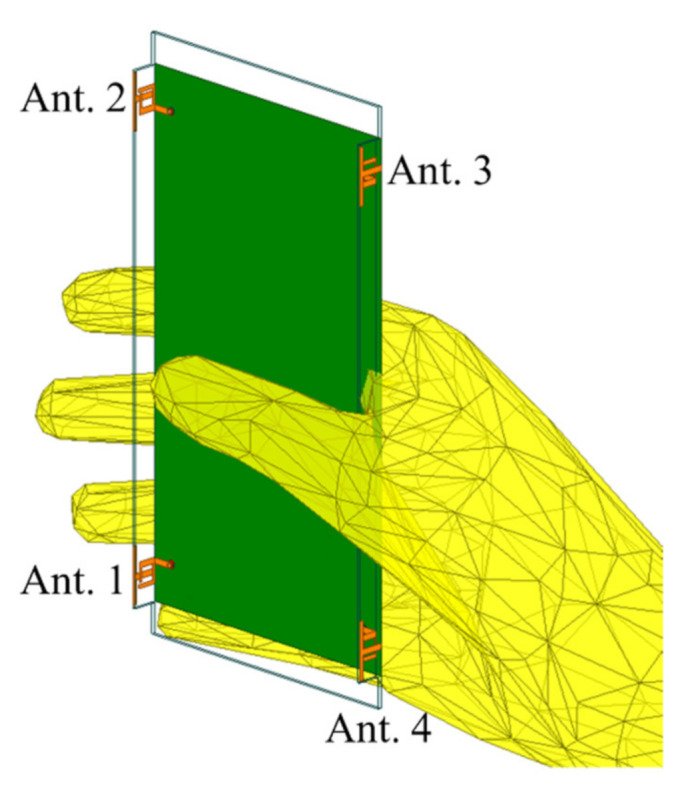
Simulated application scenario when the device is held in a single hand.

**Figure 6 micromachines-12-00489-f006:**
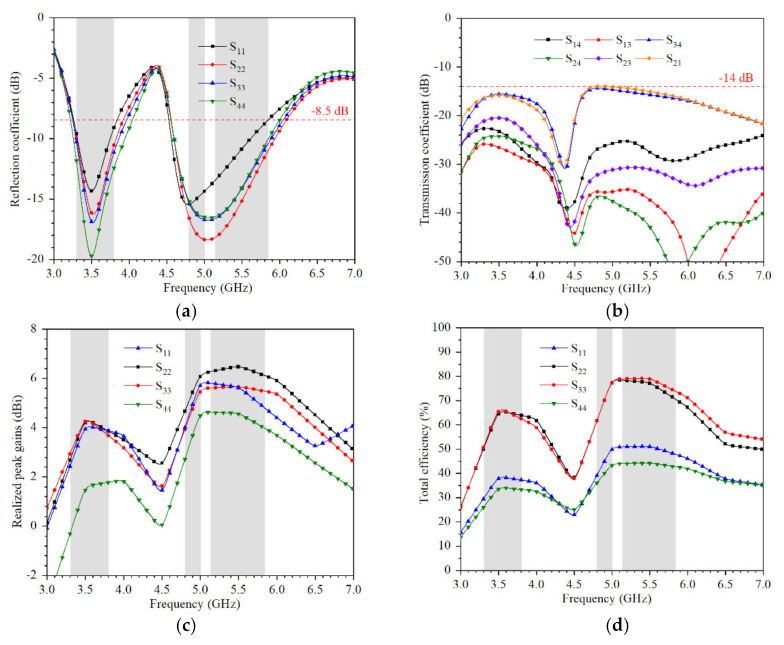
Simulated coefficients when the device is held in single hand: (**a**) reflection coefficient; (**b**) transmission coefficient; (**c**) realized peak gains; (**d**) total efficiency.

**Figure 7 micromachines-12-00489-f007:**
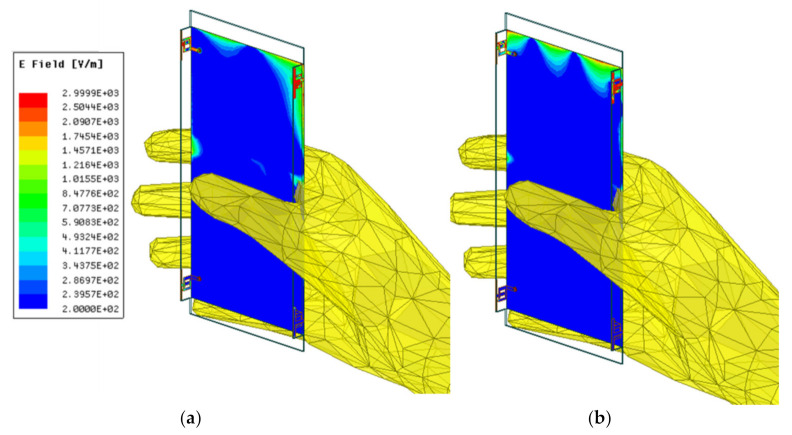
Simulated electric field distributions when the device is held in single hand at: (**a**) 3.5 GHz; (**b**) 5.5 GHz.

**Figure 8 micromachines-12-00489-f008:**
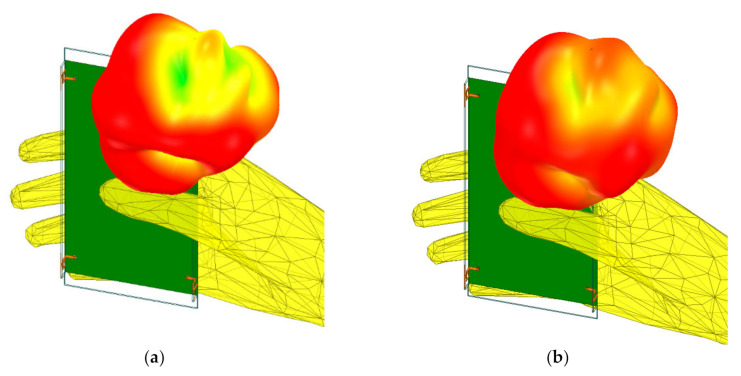
Simulated 3D radiation patterns when the device is held in single hand at: (**a**) 3.5 GHz; (**b**) 5.5 GHz.

**Figure 9 micromachines-12-00489-f009:**
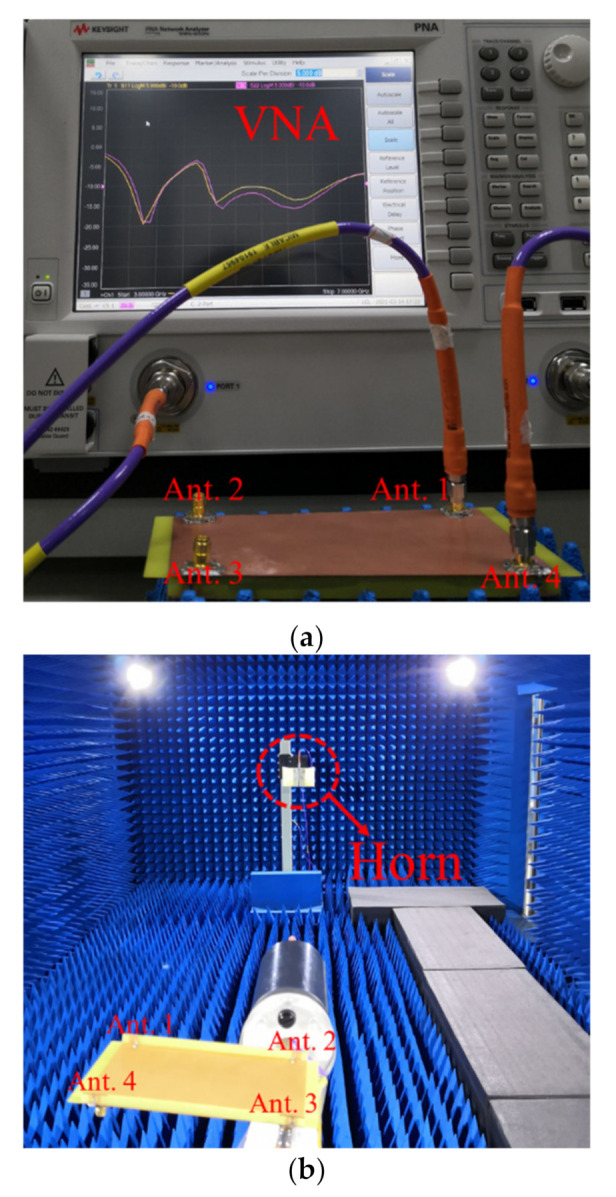
Photo of measurement experiment. (**a**) Vector network analyzer (VNA) instrument; (**b**) microwave anechoic chamber; (**c**) fabricated antenna prototype.

**Figure 10 micromachines-12-00489-f010:**
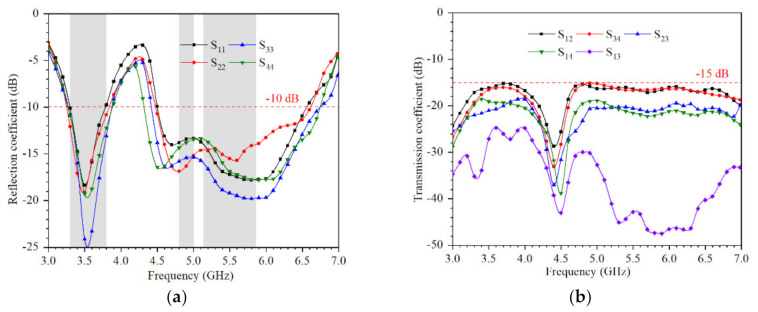
Measured coefficients of the proposed 4 × 4 MIMO system: (**a**) reflection coefficient; (**b**) transmission coefficient.

**Figure 11 micromachines-12-00489-f011:**
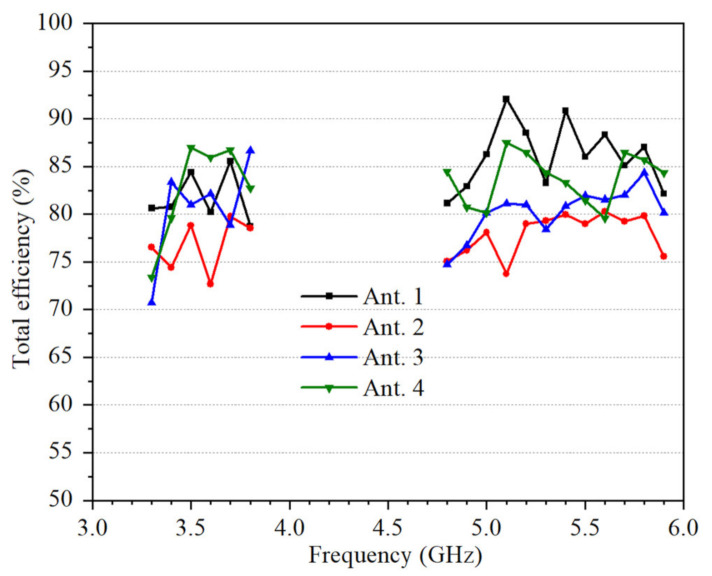
Measured total efficiencies of the proposed 4 × 4 MIMO system.

**Figure 12 micromachines-12-00489-f012:**
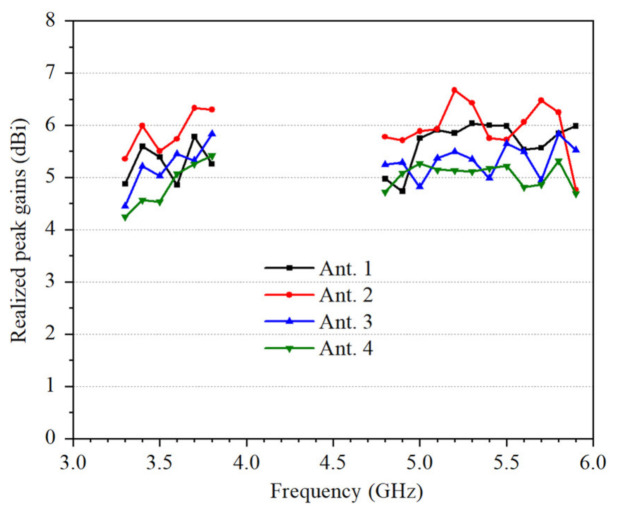
Measured realized peak gains of the proposed 4 × 4 MIMO system.

**Figure 13 micromachines-12-00489-f013:**
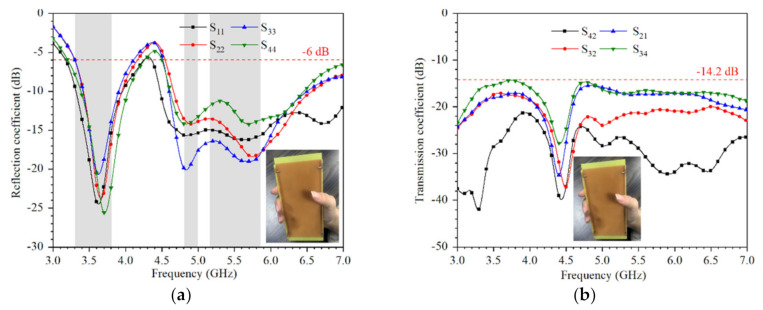
Measured coefficients when the device is held in single hand: (**a**) reflection coefficient; (**b**) transmission coefficient.

**Figure 14 micromachines-12-00489-f014:**
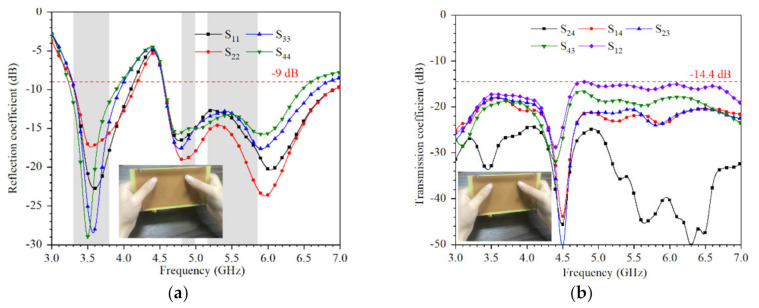
Measured coefficients when the device is held in dual hands: (**a**) reflection coefficient; (**b**) transmission coefficient.

**Figure 15 micromachines-12-00489-f015:**
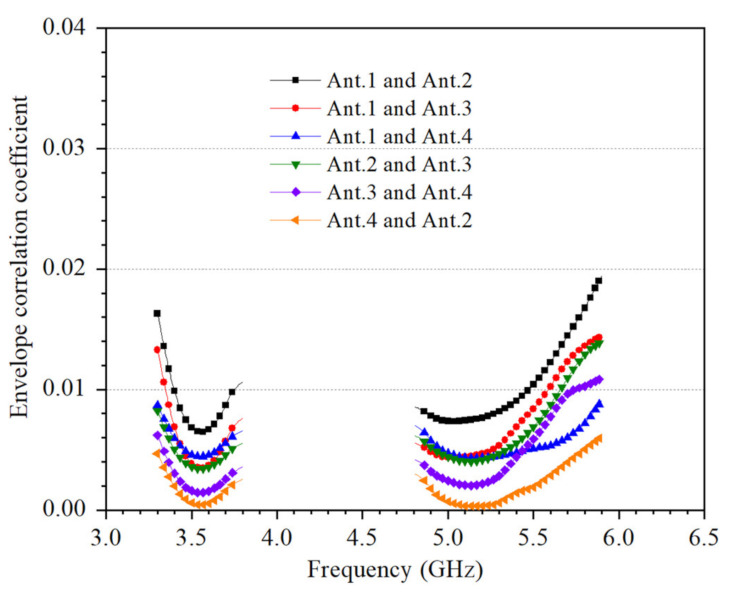
Calculated ECC of the proposed 4 × 4 MIMO system.

**Figure 16 micromachines-12-00489-f016:**
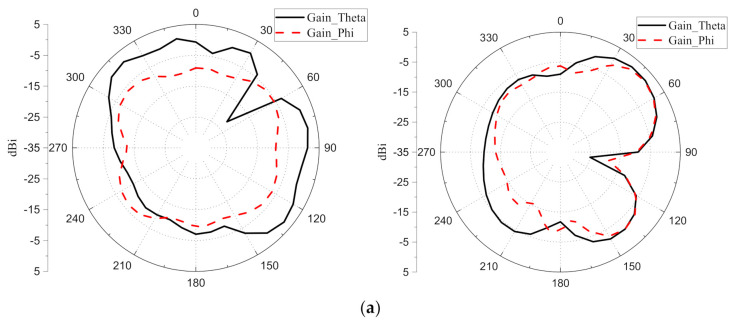
Radiation patterns of the proposed 4 × 4 MIMO system: (**a**) 3.5 GHz; (**b**) 5.5 GHz.

**Table 1 micromachines-12-00489-t001:** Comparison of the 5G MIMO smartphone antennas.

Reference	Operating Band (GHz)	Isolation (dB)	ECC	Efficiency (%)	Size (mm^3^)
[3]	2.45–2.65 3.4–3.75 5.6–6	11	<0.01	40–65 50–70 60–80	20.5 × 6.5 × 0.8
[4]	3.3–6	11	<0.1	40–71	15 × 6 × 3
[10]	3.3–3.6 4.8–5.0	10	<0.15	>60	7 × 15.5 × 0.8
[11]	3.3–6.0	12	<0.11	>50	9 × 7 × 2
[13]	3.4–3.6 4.8–5	16.5	<0.02	>70	14.9 × 7 × 0.8
[18]	3.4–3.6	17	<0.1	58	25 × 3.5 × 0.8
Proposed	3.3–3.84 4.61–5.91	15	<0.02	>70	14.9 × 7 × 0.8

## Data Availability

Data is contained within the article.

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
