# Peer review of "Dual-Band MIMO Antenna for 5G/WLAN Mobile Terminals"

_micromachines, 2021, doi:10.3390/mi12050489_

Round 1

Reviewer 1 Report

The manuscript is well-written and the presentation of results is good. However there are some concerns, 

1) the designed antenna size is compact (0.27 lambda × 0.13 lambda) which is good, but when integrated into MIMO assembly of 4 elements the size of PCB is 150 mm × 75 mm × 0.8 mm, which is huge for cellular applications. Usually, a size of 50mm × 30mm is taken for a mobile handset as a designated area to place an antenna as other components such as the battery takes the major part of the area. Authors need to give thought to this. Either change the application to make it use for indoor base-stations/mobile terminals or reduce the size to fit with the handsets. Also, explain why authors choose these big dimensions? As there are other ways to control the isolation/mutual coupling within limits in small dimensions. 

2) The draft suggests that the antenna operates on 5.15-5.35 GHz and 5.725-5.85 GHz bands too, but radiation efficiency and gain of these bands is not provided. 

3) The antenna design looks very similar to a conventional monopole-loop antenna. Authors are suggested to compare it with similar design geometries in terms of shape and performance to highlight the novel aspects in their designed antenna.

4) Fig.1 (a) and fig. 5 are very unclear. It is difficult to visualize what is present in which layer in a 3D structure. 

4) Check for grammar and typos. 

Reviewer 2 Report

This paper presents a dual-band four-element MIMO antenna for 5G mobile communication. The simulation and measurement are in good agreement. The technical merit is valuable to be published after addressing the following concerns.

The major concern is that the novelty of this antenna array design is not clear to the reader. What’s the major improvement of the design compared to these in the literature? Need to clearly discuss them in the design section.

Line 46: The reference should be [22].

Figure 2: Please add (a) and (b) in the figure, and also discuss the different performance of the surface current and electric field distributions at two frequencies.

What are the exact values of L1 and H3 in your fabrication?

Figure 8: it will be helpful to include how the envelope correlation coefficient (ECC) is derived from the antenna pattern. Are they directly output from the HFSS simulation?

What’s the difference between Figure 7 and Figure 10? They use the same captions. Also, in the two figures, there are red ellipses and arrows. Please explain them in the text, otherwise, just remove them.

In Figure 9, please identify the difference between the plots on the left and right.

In the bottom right plot of Figure 9, why is cross-pol larger than co-pol in some azimuthal angles?

In Table 1, [4] and [13] are operating at different frequencies, so it might be unnecessary to include them in the table.

Reviewer 3 Report

It is the reviewer's view that the paper is not worth to be published due to the following reasons:

  1. The wavelength of the structure ranges between 36 and 75mm. For an antenna array, the element spacing is typically in the range of Lambda/2 which is not the case here.
  2. The authors state that the antenna configuration can be used for handheld devices. When using such a device, it is usually carried in the persons hand, so that the persons hand needs to be taken into account for the simulation. Neglect can cause the antenna to fail.
  3. The authors state in line 122 that due to its symmetric structure, only S1X values need to be investigated. This statement is wrong since feeding only one of 4 antennas will cause a unsymmetric behavior.
  4. The authors also state, that the frequency shift between simulation and measurement is due to several factors such as SMA-connector losses, but the losses will only tend to a reduction in the antennas gain and not in a shift in its resonant frequency
  5. The paper would need fundamental revision in both grammar and spelling.

Round 2

Reviewer 1 Report

The authors have addressed all the raised comments and made a significant improvement to the manuscript. The draft is in good shape now. The draft in the present form is a good scientific contribution.

Author Response

I really appereciate your kindly support.

Reviewer 3 Report

The revision has significantly improved the quality of the paper. It is the reviewer's view that still some improvements are necessary:

  1. The authors state that they included SAR measurements. In fact only S-parameter measurements are added. The S-parameter change significantly when touching the area of one antenna with the hand. Real SAR measurements or at least radiation measurements would be interesting here.
  2. The authors present the S-Parameter measurements of 4 identical antennas in Fig 6a). In fact all antennas differ significantly from each other. Is this only due to manufacturing tolerances?
  3. The authors show the crosstalk between the antennas in Fig 6b). Due to the symmetrical structure S21 and S34 should be identical, but differ for about 10dB! Why does this happen?

Author Response

Thank you very much for your comments, please see the attachment.
